# Prevalence of Human Papillomavirus (HPV) Infection and the Association with Survival in Saudi Patients with Head and Neck Squamous Cell Carcinoma

**DOI:** 10.3390/cancers11060820

**Published:** 2019-06-13

**Authors:** Ghazi Alsbeih, Najla Al-Harbi, Sara Bin Judia, Wejdan Al-Qahtani, Hatim Khoja, Medhat El-Sebaie, Asma Tulbah

**Affiliations:** 1Department of Biomedical Physics, King Faisal Specialist Hospital and Research Centre, Riyadh 11211, Saudi Arabia; nharbi@kfshrc.edu.sa (N.A.-H.); sbinjudia55@kfshrc.edu.sa (S.B.J.); walqahtani1@ksu.edu.sa (W.A.-Q.); 2Department of Histology, King Saud University, Riyadh 11451, Saudi Arabia; 3Department of Pathology & Laboratory Medicine, King Faisal Specialist Hospital and Research Centre, Riyadh 11211, Saudi Arabia; hkhoja@kfshrc.edu.sa (H.K.); tulbah@kfshrc.edu.sa (A.T.); 4Department of Radiation Oncology, King Faisal Specialist Hospital and Research Centre, Riyadh 11211, Saudi Arabia; melsebaie@gmail.com

**Keywords:** head and neck cancer, squamous cell carcinoma, human papillomavirus (HPV), oral cavity tumors, oropharyngeal cancer, p16Ink4a biomarker, p16-immunostaining, prognosis, overall survival

## Abstract

Head and neck squamous cell carcinoma (HNSCC) shows wide disparities, association with human papillomavirus (HPV) infection, and prognosis. We aimed at determining HPV prevalence, and its prognostic association with overall survival (OS) in Saudi HNSCC patients. The study included 285 oropharyngeal and oral-cavity HNSCC patients. HPV was detected using HPV Linear-Array and RealLine HPV-HCR. In addition, p16INK4a (p16) protein overexpression was evaluated in 50 representative cases. Oropharyngeal cancers were infrequent (10%) compared to oral-cavity cancers (90%) with no gender differences. Overall, HPV-DNA was positive in 10 HNSCC cases (3.5%), mostly oropharyngeal (21%). However, p16 expression was positive in 21 cases of the 50 studied (42%) and showed significantly higher OS (*p* = 0.02). Kaplan–Meier univariate analysis showed significant associations between patients’ OS and age (*p* < 0.001), smoking (*p* = 0.02), and tumor stage (*p* < 0.001). A Cox proportional hazard multivariate analysis confirmed the significant associations with age, tumor stage, and also treatment (*p* < 0.01). In conclusion, HPV-DNA prevalence was significantly lower in our HNSCC patients than worldwide 32–36% estimates (*p* ≤ 0.001). Although infrequent, oropharyngeal cancer increased over years and showed 21% HPV-DNA positivity, which is close to the worldwide 36–46% estimates (*p* = 0.16). Besides age, smoking, tumor stage, and treatment, HPV/p16 status was an important determinant of patients’ survival. The HPV and/or p16 positivity patients had a better OS than HPV/p16 double-negative patients (*p* = 0.05). Thus, HPV/p16 status helps improve prognosis by distinguishing between the more favorable p16/HPV positive and the less favorable double-negative tumors.

## 1. Introduction

Head and neck (H&N) cancer is the 9th most frequent malignancy worldwide, accounting for around 5% of all new cases and exhibits wide demographic variations [1,2]. The global incidence is estimated at 600,000 cases per year, with evidence indicating rising trends especially in young adults [3]. Squamous cell carcinoma (SCC) is the most common type of Head and Neck (H&N) cancers [4], and accounts for more than 90%. Lifestyle and several anecdotal risk factors are suspected to contribute to the development of various H&N cancers, including smoking, chewing (smokeless) tobacco and other products, alcohol consumption, dietary factors, chemical irritants, and poor oral hygiene [5]. Recently, however, infection with human papillomavirus (HPV) was recognized as being an important determinant and independent of other risk factors for H&N squamous cell carcinoma (HNSCC) [6,7,8]. While infection with the Epstein–Barr virus (EBV) is a known risk factor for nasopharyngeal carcinoma, HPV is mostly linked to a subset of HNSCC, particularly oropharyngeal cancer [9,10].

Syrjanen et al. [11] were historically, the first to evoke such an association between HNSCC and HPV based on histopathological observations followed by confirmation of HPV DNA presence in oral lesions [12]. These results gained momentum when it was later observed that the incidence of oropharyngeal squamous cell carcinomas in young patients (<50 years old), particularly in the tonsils and the base of tongue, increased significantly even though most patients are not regular tobacco or alcohol users [9,13]. This observation holds true despite the initial decline in H&N cancer incidence in North America consequent to active antismoking campaigns [9,14,15]. Large meta-analysis studies have estimated that 32% of HNSCC are associated with HPV, with higher rates in oropharyngeal than oral cancers [16]. Globally, HPV16 was considerably the most common subtype, accounting for 82% of all HPV positive cases, followed by HPV18 and ensued by a minority of other sporadic genotypes.

The more prominent turning point is that HNSCC HPV-positive cancers appear to form a distinct tumor entity from smoking- and alcohol-related counterparts with distinguished epidemiology, genetics, characteristic histopathology, therapeutic response, and predictive clinical outcome to chemo-radiation treatment [17,18,19]. Their noticeable molecular characteristics include p16Ink4a (p16) overexpression, modulation of PI3K/AKT and Wnt pathways, and lack of inactivating p53 mutations [20]. Furthermore, the observed HPV-associated overexpression of p16 protein in HNSCC has been largely considered as a surrogate marker diagnostic for HPV infection and also prognostic for a more favorable treatment outcome [21]. It was incorporated in the recent release of TNM-8, leading to marked changes in the classification of these malignant tumors [17,22]. Thus, the detection of HPV infection and histopathological determination of p16 protein expression in tumor samples are expected to gain importance in clinical settings and marks a major shift in managing HNSCC cancer patients.

The epidemiology of HPV infection is known to have wide variations in human populations, remnant of socioeconomic, ethnic, and genetic predisposing factors [23,24]. According to the Saudi Cancer Registry, H&N cancers, excluding nasopharynx, forms about 4% of all malignancies in this country [25]. If a third of those tumors are HPV-driven, then the projected burden of HPV, along with cervix, uterine, and other anogenital malignancies, would represent, in both genders, approximately 3% of all cancers in Saudi Arabia [26]. This is a significant medical issue for a health authority, particularly for the cost-effectiveness analysis of implementing a nationwide HPV vaccine in order to render these HPV-mediated tumors preventable. This is in addition to introducing personalized treatment modalities to boost cure rate and reduce patients’ morbidity and mortality. However, actual data about the implication of HPV infection in HNSCC in Saudi cancer patients is completely lacking. Therefore, the main aims of this retrospective exploratory study were to determine the prevalence of HPV infection and its oncogenic genotypes, and the association with patients’ overall survival (OS). The correlation with p16 protein expression was also studied in a subset of these tumors, to assess the prognostic values of HPV status and p16 protein positivity.

## 2. Results

### 2.1. Patients and Clinical Data

The characteristics of the 285 H&N cancer patients included in the study are summarized in Table 1. The age of patients at diagnosis of HNSCC ranged between 22 and 90 years (median = 57 years). The incidence showed a Gaussian distribution that increased with age to reach a peak at 59-year-old, and then decreased gradually (Figure 1A). There were 120 females and 165 males with no noticeable difference by gender in the distribution of cancer patients by age at diagnosis. Although the median age of females (60-year, range 23–90) was slightly higher than that of males (57-year, range 22–90), there was no statistical difference (*p* = 0.09; two-tailed Mann–Whitney Rank Sum test). By anatomical sites, 28 patients (10%) had oropharyngeal while 257 (90%) had oral cavity cancers. Interestingly, the number of oropharyngeal cancer cases increased with time: 8 cases were diagnosed in 2002–2008 (142 patients) compared to 20 cases in 2009–2016 (143 patients). The distribution of sub-anatomical sites of oral and oropharyngeal cancers by 5-year age groups is illustrated in Figure 1B. The stage of the tumors varied from T1N0M0 to T4N2cM0 with 63% of patients having early stage (T1–2) compared to 37% with advanced (T3–4) tumors. Patients followed mainly standardized curative treatment according to the stage of the tumor as described above. The length of patients’ follow-up extends to 15 years (mean = 4.36 years; standard deviation = 3.88) after diagnosis. There were seven ambiguous cases with locally advanced invasion, without evidence of distal metastatic cancer, who displayed an overall short mean survival of about six months.

Patients’ characteristics were significantly associated with OS for groups of age, separated by the median of 57 years/old (*p* < 0.001), and smoking (*p* = 0.02), while gender (*p* = 0.28) had no effect (Figure 2). In addition, patients’ OS declined significantly (*p* < 0.001) from T1 to T4 (Figure 3A). Although oropharyngeal cancer displayed a slightly improved OS compared to oral cavity patients, the difference was not statistically significant (*p* = 0.14; Figure 3B). Alcohol dependence or abuse was reported in only 8% of the patients, meanwhile years of daily tobacco smoking was common in this cohort (62%) in both genders, comprising 28% who were Shamma (a chewing tobacco mixture) users.

### 2.2. Detection of HPV Infection and Genotyping

The Linear Array HPV Genotyping Test was first used to detect and genotype HPV infection. Results indicated that only 10 patients (3.5%) were HPV positive while 275 specimens (96.5%) proved to be negative after at least two separate tests and an independent concordant confirmation using the RealLine HPV HCR Genotype (Table 1). By HPV genotype, nine cases were HPV16 and one case was HPV33. These were detected in three females and seven males with a median age of 57 years (range 32–78). A Mann–Whitney Rank Sum test showed no significant difference in the median age between HPV-positive and HPV-negative patients (*p* = 0.65). By anatomical site, 21% (6/28) of the oropharyngeal and 2% (4/257) of the oral cavity cancers were positive for HPV infection. Most frequent HPV-positive cases were recorded from the tonsils (3/12), the base of the tongue (3/14), the retromolar (2/24), followed by the buccal (1/19) and the tongue (1/198). Thus, the highest HPV-DNA positivity was in oropharyngeal cancers (21%), which is not statistically different from the worldwide 36–46% estimates (*p* = 0.16). Overall, survival analyses showed a trend toward better OS for HPV-positive (67% survival) compared to HPV-negative (27% survival) patients but that did not reach statistical significance (*p* = 0.12), most probably due to the small number of HPV-positive cases (Figure 4A).

### 2.3. p16Ink4a Protein Immunohistochemical (IHC) Staining

In view of the small number of HPV-positive HNSCC patients found in this cohort, and the limited amount of pathological materials available, a subset of 50 representative specimens were processed for p16 protein IHC staining. These included all the 28 oropharyngeal cases, which are known as highly suspicious for HPV infection. In addition, 22 cases of oral cavity subsites were processed comprising 10 retromolar, 10 tongue and 2 buccal for which at least one cancer was positive for HPV. Thus, the 10 HPV-positive tumors along with 40 HPV-negative cases were included. Examples of p16 protein IHC strong (positive) and weak (negative) staining is given in Figure 5. In total, p16 was positive in 42% (21 tumors) of the 50 tested cases. Interestingly, p16 was positive in all the 10 HPV-positive tumors (double-positive for HPV and p16-over-expression) in addition to 11 HPV-negative cases (single-positive for p16 overexpression) while the remaining 29 samples were double-negative. The p16 positivity was 39% in the 28 oropharyngeal cases and 45% in the 22 cases studied of oral cavity cancers. There were no significant differences in patients’ age or male to female ratios between p16 positive and negative cases (*p* > 0.05). A survival analysis showed a statistically significant (*p* = 0.02) better OS for p16-positive (64% survival) compared to p16-negative (29% survival) patients (Figure 4B). In addition, the survival analysis of the combined HPV/p16 status in the 50 cases studied (Figure 4C), showed an overall significant difference in OS (*p* = 0.05), whereby HPV and/or p16 positive patients displayed better survival (64–67% survival) compared to HPV/p16 double-positive patients (29% survival). However, there was no significant difference between double-positive and single positive patients (*p* = 0.85).

## 3. Discussion

H&N cancer is an important health issue worldwide [1,3,4]. The identification of HPV infection as an independent risk factor, particularly in the oropharynx, with favorable prognosis for treatment response and survival spurred out research to stratify patients to deliver more personalized treatment [6,7,8]. However, HPV-associated HNSCC cancers are known to display wide epidemiological variation between populations [24]. To the best of our knowledge, this is the first study on the association between HNSCC and HPV in Saudi cancer patients. We have systematically reviewed 1633 medical records spanning more than one decade of H&N cancer patients admitted at our tertiary care hospital. We have first targeted oropharyngeal and secondly oral cavity cancers as they are potentially the most associated with HPV [27]. Following a review of the pathological samples, only 285 cases were available for the study.

Patients’ characteristic data showed that the incidence of oropharyngeal and oral cavity HNSCC increased with age from 22 years to reach a peak at 59 years, then decreased to 90 years old (Figure 1A). There were no obvious differences in the incidence by gender or cancer sites. Females composed 42% of the patients compared to 58% of males in this cohort. This relatively high incidence in females is rather unusual for HNSCC; however, it confirms a previous study in the country [28]. Although the exact reason is still unknown, it might be related to the increased habits of females consuming tobacco products (including Shamma) as 55% were smokers (68% in males). Oropharyngeal tumors, however, were infrequent and formed only 10% (28/285) of the patients while 90% (275/285) had oral cavity cancers (Table 1; Figure 1B). This 10% ratio is significantly lower (*p* ≤ 0.001) than the projected 32% computed from the estimated number of incident cases worldwide [1], suggesting lower incidence of these types of cancers in our population. Although this is a single institution study, the low rate is representative of the country because the King Faisal Specialist Hospital and Research Centre (KFSHRC) is the primary tertiary care referral hospital, which captures more than 50% of cancer patients in the Kingdom. The low incidence of oropharyngeal cancers found in this study is in agreement with the national registry with an age-standardized rate of 0.07 (Cancer Today, Globocan 2018 statistics on oropharyngeal cancer in Saudi Arabia available at: http://gco.iarc.fr/today/home). Nevertheless, our data indicate an increase in the number of oropharyngeal cancer cases over time as it doubled in the last decade in the country. The subsites of H&N cancers that are most frequently associated with HPV infection are the tonsils, the soft palate, the base of the tongue (oropharynx) and the retromolar (oral cavity). These together formed 18% (52/285) of the cases in this cohort. The remaining 233 patients had other various oral cavity tumors including mostly tongue (Table 1).

As expected, tumor stage had independently major impact on patients’ OS which significantly (*p* < 0.001) decreased gradually from T1 to T4 (Figure 3A). This statement incorporates the standardized patient treatment that depended mainly on tumor stage. Interestingly, our results showed that age and tobacco consumption affect prognosis, as statistically significant better survival was observed for younger-age (*p* < 0.001) and non-smokers (*p* = 0.02), but not for patients’ gender (Figure 2). As for HPV oncogenic risk factor, only 10 samples were positive for HPV-DNA out of the 285 patients (Table 1). This indicates that only 3.5% (10/285) of HNSCC are infected with HPV in our cancer patients. In an early global systematic meta-analysis that comprised 60 eligible studies and included 5046 cases, the overall HPV prevalence in HNSCC was 25.9% with significantly higher presence in oropharyngeal (35.6%) than in oral (23.5%) and in laryngeal (24%) cancers [29]. In a more recent meta-analysis, HPV-DNA was detected in 32% (3837 out of 12,163 cases) of HNSCC, with again a higher prevalence in oropharyngeal (46%) than oral (24%) or laryngeal and hypopharyngeal (22%) cancers [16]. The 3.5% prevalence of HPV in HNSCC in our patients is significantly lower than the estimated 32–36% worldwide (*p* ≤ 0.001, one sample *z*-test).

In agreement with published data, the highest prevalence of HPV was observed in oropharyngeal cancers (21%), mostly in tonsils (3/12) followed by the base of the tongue (3/14). Although this 21% prevalence remains below the compiled worldwide estimates of 36–46%, it was not statistically different (*p* = 0.16), indicating similar pathogenic association. As for oral cavity, only 2% were HPV-positive, mainly recorded from the retromolar (2 cases), the tongue, and the buccal cavity (one case each). This low prevalence is in agreement with a recent study by Vidal Loustau et al. [30], but again much lower than the worldwide estimate of 23.5% stated above. Overall, these results imply that the prevalence of HPV-driven HNSCC in our population is very low. The reasons for this low rate is unknown, but could be related to the predominance of other risk factors, such as various tobacco products consumption, which is as high as 62% of patients, socio-cultural differences, or the presence of relative protective variants of genetic predisposing factors as has been shown previously for cervical cancer patients [31]. Most infections (90%) were with HPV16 (9/10) followed by 10% with HPV33 (1/10) genotypes. These results are in line with other studies even though HPV33 was much less commonly observed. In fact, the latter was a case of SCC of the tongue (Table 1). The patient was a young male who had bone marrow transplant for leukemia two decades ago. It is probable that his relatively compromised immune response resulted in a persistent HPV infection with this rare HPV33 genotype leading to this neoplasia [32].

One of the most significant advancements in H&N oncology of the precedent decade is the demonstration that cancer patients with HPV-mediated HNSCC, particularly in the oropharynx, have p16 protein expression and are associated with significantly improved treatment outcomes expressed as higher rate of patients’ survival, compared to HPV-negative patients [6,33]. Furthermore, these observations have laid the foundations for exploratory clinical trials examining the impact of proposed “treatment deintensification” for patients with HPV-driven cancers [34,35]. The rationale is to improve treatment outcome, by reducing side effects without compromising tumor control. Our results for p16 protein expression in 50 HNSCC cases showed that 42% (21/50) were positive for p16 over-expression (Figure 5, sample A), including all the 10 HPV-positive tumors (double-positive). Interestingly, 11 p16-positive cases were HPV-negative (single-positive). It is known that the clinical relevance of HPV-DNA positivity is a matter of debate, because it is likely to represent both transcriptionally active (RNA+) and inactive (RNA−) HPV genomes. Therefore, detection of HPV-RNA by in situ hybridization is considered the gold standard for clinically relevant, HPV transcriptionally active lesions. However, the availability of this RNA methodology and concern for lower sensitivity compared to the affordable HPV-DNA polymerase chain reaction (PCR) led to the evaluation of IHC-p16 protein as a surrogate marker for the presence of active HPV in tumor cells [15]. Therefore, the Union for International Cancer Control (UICC) 8th edition defines HPV-mediated oropharyngeal cancer by use of p16 immunohistochemistry [36]. With this argument in mind, it is probable that the p16-positive samples in the 50 cases studied represent active HPV infection, as many of them could not be picked by the HPV-DNA PCR-based techniques. In such an arguable case, it is acknowledged that the incidence of HPV-related HNSCC would be higher (up to 42%) in our patients. However, a larger study with more patients is needed to confirm this assumption, particularly that active HPV-RNA is considered rare in non-oropharyngeal tumors [37]. The alternative view could be that it is possible that p16 positivity is not exclusively related to HPV infection, which would debate its use as a surrogate marker for the presence of HPV in all HNSCC [38]. Indeed, discrepancies in the p16/HPV-positivity have been observed and it is questionable if all HPV-positive and/or p16-positive tested cancers are HPV-driven. It is possible that sometimes HPV is an innocent bystander and p16 is independently positive [39]. This highlights the importance of identifying robust fingerprints of HPV-driven carcinogenesis to improve the estimate of HPV-attributable HNSCC and to predict the effectiveness of implementing preventive HPV vaccination and therapeutic interventions.

The relationship between HPV and OS after the treatment showed a clear trend toward a longer survival of HPV-positive patients (Figure 4A) as described elsewhere [40]. However, in our study a survival analysis did not reach statistical significance (*p* = 0.12) due to, most probably, the small number of HPV-positive cases (10/285). In addition, those patients had mainly T1–2 tumor stages with basically a favorable survival prognosis. Nonetheless, in a subset of 50 cases, a statistically significant (*p* = 0.02) better survival was observed for p16-positive compared to p16-negative patients (Figure 4B). Although tumor stages were distributed more evenly in this group (T1–2 = 58%, T3–4 = 42%), there was slight preponderance (32%) of early stages in the p16-positive patients and vice versa. Furthermore, tumor stage as well as age remained significantly associated with OS in multivariate analysis (Table 2). In addition, the treatment offered to patients showed statistically significant (*p* = 0.006) association with OS. It also shows that surgery, which mainly underlies early stage tumors, result in higher survival compared to any other combined treatment. In other word, this result essentially captures that of the tumor stage since the treatment was stage standardized with some subtle adaptation to each individual case. In addition, a trend toward association with OS was apparent for HPV/p16 and smoking status but they did not reach statistical significance in the multivariate analysis. Nevertheless, our results are in overall agreement with published data with the overwhelming belief that p16-positive HNSCC have improved locoregional tumor control and survival with conventional therapy [21,27,41]. Potential future refinement could be brought about by including the copy number variation of the CDKN2A gene that encodes p16ink4a [42], and involving other related prognostic biomarkers such as epidermal growth factor receptor (EGFR), and key transcription factors as molecular signature of HPV presence [19,43], especially for de-escalation of radiotherapy combined with anti-EGFR receptor treatment [44].

While the study points out toward the need of systematic testing of p16 overexpression, results obtained in a subset of patients, the results are also in line with a recent study evaluating the 8th TNM classification that integrates p16 status (as independent or surrogate markers for HPV infection) in oropharyngeal cancer [17,45]. The study included 1204 patients where 32% were p16-positive which is close to the 42% observed in our study despite the limited number of cases processed for p16 expression. Importantly, the authors found that 12% of p16 positive cases were negative for HPV-DNA. This HPV-negative subgroup had distinct features and a poorer OS. Therefore, we have analyzed the OS with the various combination of HPV and p16 status in a subset of 50 cases with sufficient pathological materials (Figure 4C). Interestingly, the Kaplan–Meier Log-Rank survival analysis showed a significant difference (*p* = 0.05) where HPV/p16-positive cases showed substantially better OS than double-negative patients. Although double-positive cases showed slightly better survival than single-positive patients, the difference was not statistically significant, most probably due to the small number of patients who tested positive for HPV and/or p16. Nonetheless, taken together, these results highlight the importance of performing independent HPV and p16 testing when predicting individual patient’s prognosis [39,46]. These results are in line with a recent study on oropharyngeal cancer in four Catalonian hospitals where double positivity for HPV-DNA/p16 showed the strongest diagnostic biomarker accuracy and prognostic value [47]. The findings may have major impact in clinical practice, in particular when selecting cases for deintensified treatment regimens.

## 4. Materials and Methods

### 4.1. Ethical Considerations

The study was carried out using archival pathologic materials of H&N cancer obtained during routine diagnostic procedures. The samples were anonymized and processed with no patients’ identifiable characters. The study was reviewed by the institutional review board and approved by the Research Centre Ethics Committee at the King Faisal Specialist Hospital and Research Centre (KFSHRC) under the number RAC#2130 025.

### 4.2. Clinical Specimens

Medical records of 1633 H&N cancer patients diagnosed between 2002 and 2016 at the KFSHRC tertiary care hospital were screened. The main eligibility criteria were adult patients with squamous cell carcinoma in anatomical location potentially associated with HPV infection. Following the exclusion of palliative cases and cancer sites that had not been proven to be HPV-driven (for instance nasopharynx, salivary glands, and trachea), only 330 patients with oropharyngeal and oral cavity tumors remained for possible inclusion. After the examination of the histopathological slides, 285 patients’ samples were included due to the limited amount of pathological blocks available for this study. Patients’ treatment with curative intent followed timely standard clinical guidelines that depends on primary tumor location and extension [48]. Briefly, early stage (I–II) oral cavity tumors were treated with conservative surgery (S) and/or external radiotherapy (RT) 66–70 Gy in 33–35 fractions. Locally advanced stages (III–IV) were treated with surgery including reconstruction plus postoperative radiotherapy 60–66 Gy in 30–33 fractions. Patients found at surgery to have high-risk features were treated with post-operative chemoradiotherapy (S + CRT) 66 Gy in 33 fractions with 3 weekly cisplatin 100 mg/m^2^. Patients having resectable tumors with poor prognosis were treated with combined concomitant CRT 66–70 Gy in 33–35 fractions with 3 weekly cisplatin 100 mg/m^2^. A combined concomitant CRT was also the standard treatment in oropharyngeal and non-resectable oral cavity cancer patients. Cetuximab was used for patients who were not fit for cisplatin chemotherapy. Radiotherapy modalities included 3D conformal that was gradually replaced with intensity-modulated radiation therapy (IMRT) in 2006, and also RapidArc in 2010 and TomoTherapy in 2012. Although some HPV-related histopathologic features were available for few cases, the treatment followed the same guideline for all patients with no difference between positive and negative HPV.

### 4.3. DNA Extraction

Formalin-fixed, paraffin-embedded (FFPE) tissues proven to contain tumor sections of the 285 patients were obtained from the pathology department’s archive. For each case, 3–6 sections of 10 μm thickness were taken from the block for the extraction of DNA using the QIAamp DNA FFPE tissue kit (Qiagen, Dusseldorf, Germany), using the manufacturer’s recommended instructions. Briefly, the FFPE sections were deparaffinized using xylene followed by ethanol to extract residual xylene. The specimens are covered with ATL lysis buffer with 20 μL proteinase K (20 mg/mL, Roche, Mannheim, Germany) and incubated at 56 °C and 90 °C for 1 h each. Then, 2 μL 100 mg/mL DNase-free RNase A (Qiagen) was added, mixed and incubated at room temperature for 2 min. After the lysis and heating, followed by binding and washing steps, DNA was eluted in 50 μL of ATE buffer and quantified using a NanoDrop 2000c Spectrophotometer (Thermo Fisher Scientific, Waltham, MA, USA).

### 4.4. HPV Detection and Genotyping

Two different methods were consecutively used to detect and genotype HPV infection in all the H&N samples along with HPV negative (HTB-31) and HPV-16 positive (HTB-35) external controls:

1 The Linear Array HPV Genotyping Test (LA HPV GT; Roche Diagnostics, Mannheim, Germany). This PCR-based test detects and genotypes the 37 most common anogenital HPVs (13 high-risk: 16, 18, 31, 33, 35, 39, 45, 51, 52, 56, 58, 59, 68, and 24 low-risk: 6, 11, 26, 40, 42, 53, 54, 55, 61, 62, 64, 66, 67, 69, 70, 71, 72, 73 (MM9), 81, 82 (MM4), 83 (MM7), 84 (MM8), 89 (CP6108) and IS39). Procedures followed the manufacture’s instruction described in detail previously [31,49]. Briefly, the methodology involves the PCR amplification of the target DNA, the hybridization of the amplified DNA segments to oligonucleotide probes immobilized on strips of membranes, and finally, the colorimetric detection of the hybridized products using the Linear Array Detection Kit. The adequacy of samples is determined by the β-globin gene as an internal control. HPV positive reactions show visible blue bands localized on the strip. The HPV genotype is determined using the HPV reference guide provided in the kit. Results were deemed negative when no HPV band was detected after at least 2 independent tests with confirmed adequacy of samples.

2 RealLine HPV HCR Genotype Fla-Format (Bioron, Diagnostics GmbH, Ludwigshafen, Germany). This Real-Time PCR test allows the differential determination of the 12 most frequent high-risk HPV-DNA genotypes, 16, 18, 31, 33, 35, 39, 45, 51, 52, 56, 58 and 59, isolated from clinical specimens. It is based on the detection of the unquenched fluorescence produced by a specific reporter molecule that intensifies as PCR reaction cycles increased. The reporter molecule is a fluorophore-quencher dual-labeled DNA-probe designed to bind exclusively to the HPV-DNA target region. Fluorescent signal increases as a result of the cleavage of the probe by Taq DNA-polymerase exonuclease activity, which separates the fluorescent dye from the quencher during the repeated cycles of hybridization and amplification. The threshold cycle value (Ct) is defined as the cycle number at which the generated fluorescence crosses a set threshold within the reaction where the signal increases significantly above the background fluorescence of the procedure. Ct depends on the initial quantity of the HPV-DNA template present. A positive HPV control is run with the samples and an internal control (IC) detecting the content of human DNA (β-actin) is used to validate the quality of sampling and improve the reliability of results by preventing generation of false negatives which can be caused by the possible loss of a DNA template during sample preparation.

3 Procedures followed the manufacture’s recommended methodology. Briefly, to analyze each sample for the detection of the 12 HPV-DNA genotypes, 4 tubes containing Master Mix (MM1, MM2, MM3, MM4) in 0.2 mL 96-well plates were used. The amplification is carried out on the CFX96 Touch Real-Time PCR Detection System (Bio-Rad, Hercules, CA, USA) using the recommended cycling program. The sample is flagged as positive (i.e., containing HPV-DNA) when the Ct value via the fluorescent dyes, FAM, HEX, and ROX channels, for this sample (in any of MM 1–4 tubes) is less than or equal to 35 for HPV types 31, 33, 35, 39, 45, 51, 52, 56, 58, 59, or is less than or equal to 40 for HPV types 16 and 18. The HPV genotype is determined using a reference table provided by the manufacturer, which correlates each MM with an individual dye channel to one of the 12 specific high-risk HPV types.

### 4.5. Immunohistochemical (IHC) Staining of p16 Protein Expression

Procedures examining the expression of p16 protein were carried out using a Bond-III Automated IHC/ISH Stainer (Leica Biosystem, Wetzlar, Germany) according to manufacturer’s instruction and reagents. Briefly, where available, 4 μm FFPE sections were mounted on glass microscope slides coated with Poly-L-Lysine. They were deparaffinized using Bond Dewax Solution (Leica Biosystem), rehydrated, and washed with Bond Wash Solution. The slides were incubated with Bond Epitope Retrieval Solution and heated at 100 °C for 20 min, washed, and Peroxide Wash Solution applied for 5 min. The p16 primary antibody (mouse monoclonal Anti-p16INK4a (E6H4), Ventana, Tucson, AZ, USA) was added on the slides for 15 min, followed by the anti-mouse secondary antibody (Post Primary Rabbit anti mouse IgG, ProClin, Leica Biosystem) for 8 min and the Bond Polymer Refine Detection solutions with intermittent washing. Slides were counterstained with Hematoxylin, and then dehydrated and mounted with DPX by using a Tissue-Tek film coverslipper (Sakura Finetek, Tokyo, Japan). Negative controls were obtained by excluding the primary antibody. Scoring of p16 IHC cytoplasmic and nucleic staining were evaluated by an experienced pathologist, based on defined characteristics whereby p16 was scored as positive if it was strong and diffuse (>70% of tumor cells), and negative if absent, weak, or focal [50].

### 4.6. Statistical Analysis

A one sample z-test was used to detect differences in proportions when the referenced proportion was deemed constant. The non-parametric Mann–Whitney Rank Sum test was used to assess differences between groups. A univariate Kaplan–Meier Log-Rank survival analysis was used to evaluate the relationship between various risk factors and overall survival (OS) represented by the length of patients’ follow-up. A multivariate Cox proportional hazards model was used to test the effects of multiple covariates on patients’ OS. All statistical tests conducted were two-sided. A *p*-value < 0.05 was considered statistically significant. Statistical analyses were done using the SigmaPlot platform (Version 12.5, SPSS Science, San Jose, CA, USA), and MedCalc, Ostend, Belgium (https://www.medcalc.org/calc/test_one_proportion.php).

## 5. Conclusions

This study indicates an overall low prevalence of HPV infection in our HNSCC patients. Although oropharyngeal cancer cases were infrequent, they increased over years and 21% were associated with HPV infection. Age, smoking, tumor stage, and treatment had important effect on survival. Although all HPV-positive cases were p16-positive (double-positive), the p16 positivity is not exclusive and could be positive in HPV-DNA negative tumors. HPV and/or p16 positivity had better prognosis of survival than HPV and/or p16 negative patients. An important clinical application is in the stratification of patients according to HPV and p16 status. These tests could improve survival predictions by distinguishing between the more favorable HPV-positive/p16-positive group, and the less favorable double-negative HPV/p16 group of HNSCC patients who have the worst prognosis.

## Figures and Tables

**Figure 1 cancers-11-00820-f001:**
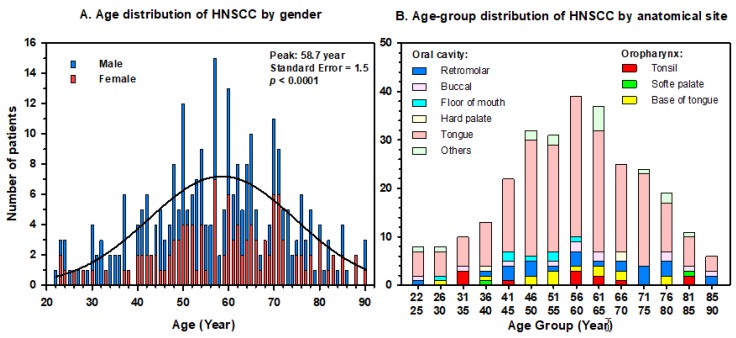
Incidence of head and neck squamous cell carcinoma in 285 Saudi cancer patients. (**A**) Age-distribution by gender of patients. Clustering analysis indicates a peak of maximum occurrence at the age of 58.7 years old. (**B**) Distribution by 5-year age groups of oropharyngeal and oral cavity tumors by sub-anatomical sites.

**Figure 2 cancers-11-00820-f002:**
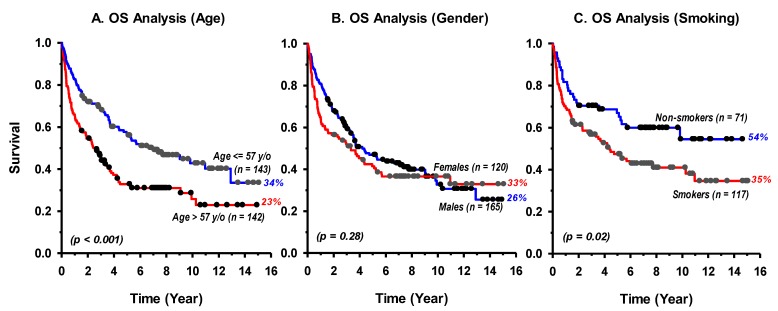
Kaplan–Meier Log-Rank overall survival (OS) analysis by patients’ characteristics of groups of age (**A**), gender (**B**), and smoking status (**C**) for 285 patients with head and neck squamous cell carcinomas.

**Figure 3 cancers-11-00820-f003:**
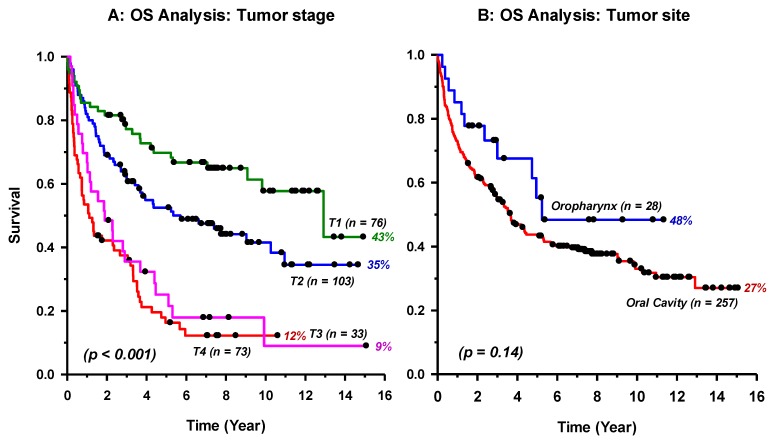
Kaplan–Meier Log-Rank overall survival (OS) analysis by tumor stage (**A**) and anatomical tumor site (**B**) of 285 patients with head and neck squamous cell carcinomas. The *p*-value in (**A**) represents the overall significance level. Al pairwise comparisons were statistically significant (*p* ≤ 0.03) except T4 vs. T3 (*p* = 0.32).

**Figure 4 cancers-11-00820-f004:**
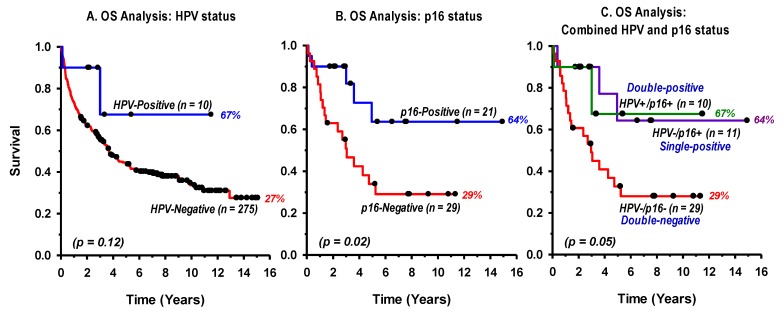
Kaplan–Meier Log-Rank overall survival (OS) analysis by HPV (**A**) status of head and neck squamous cell carcinomas in 285 cancer patients, p16INK4a (**B**), and the combination of HPV and p16INK4a (**C**) status in a subset of 50 cases. The *p*-value in (**C**) represents the overall significance level. There was no significant difference between single-positive and double-positive cases (*p* = 0.85).

**Figure 5 cancers-11-00820-f005:**
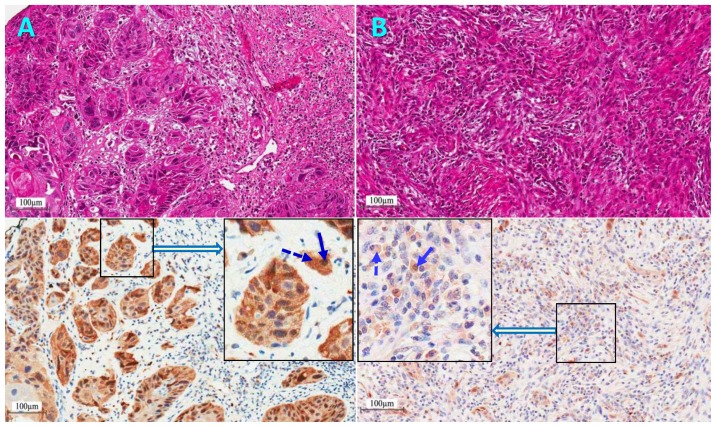
Examples of pathological tissue sections of head and neck squamous cell carcinomas stained with hematoxylin and eosin (**upper panels**) and the corresponding immunohistochemical staining for p16Ink4a (**lower panels**) showing nuclear (solid arrows) and cytoplasmic staining (dashed arrows). Sample (**A**) shows strong staining (usually involving >70% of the tumor cells) scored as p16-positive as compared to sample (**B**) with weak patchy staining, arbitrated as p16-negative.

**Table 1 cancers-11-00820-t001:** Descriptive statistics of 285 head and neck squamous cell carcinoma patients with results of HPV and p16^INK4a^ tests.

Cancer Site (ICD Code)	Number of Cases	Age Median (Range)	Gender	Smoking *	Tumor Stage	Treatment	HPV	p16
Oropharynx:								
Tonsil (C09)	12	60 (31–83)	M: 9 F: 3	Yes: 4 No: 5 N/A: 3	T1–2: 6 (N0: 2, N+: 4, M0: 6) T3–4: 6 (N0: 5, N+: 1, M0: 6)	S: 1, S+RT: 5 CRT: 4, S+RT: 2	P: 3 (HPV16) N: 9	P: 4 N: 8 N/A: 0
Soft palate (C05.1)	2	61 (37–85)	M: 2 F: 0	Yes: 1 No: 0 N/A: 1	T1–2: 0 (N0: 0, N+: 0, M0: 0) T3–4: 2 (N0: 2, N+: 0, M0:2)	S+CRT: 2	P: 0 N: 2	P: 0 N: 2 N/A: 0
Base of tongue C01	14	54 (27–78)	M: 7 F: 7	Yes: 7 No: 5 N/A: 2	T1–2: 8 (N0: 3, N+: 5, M0: 8) T3–4: 6 (N0: 0, N+: 6, M0: 6)	S: 2, S+RT: 5, S+CRT: 1 S+RT: 1, CRT: 3, S+CRT: 2	P: 3 (HPV16) N: 11	P: 7 N: 7 N/A: 0
Oral Cavity:								
Retromolar C06.2	24	62 (24–90)	M: 16 F: 8	Yes: 11 No: 8 N/A: 5	T1–2: 11 (N0: 6, N+: 5, M0: 11) T3–4: 13 (N0: 1, N+: 12, M0: 13)	S: 3, S+RT: 5, CRT: 1, S+CRT: 2 S+RT: 1, CRT: 1, S+CRT: 11	P: 2 (HPV16) N: 22	P: 5 N: 5 N/A: 14
Tongue C02	198	57 (22–90)	M: 113 F: 85	Yes: 82 No: 45 N/A: 71	T1–2: 134 (N0: 84, N+: 51, M0: 134) T3–4: 64 (N0: 24, N+: 39, M0: 64)	S: 62, S+RT: 56, S+CRT: 16 CRT: 1, S+RT: 13, S+CRT: 50	P: 1 (HPV33) N: 197	P: 4 N: 6 N/A: 188
Buccal (C06)	19	62 (24–90)	M: 9 F: 10	Yes: 7 No: 3 N/A: 9	T1–2: 11 (N0: 6, N+: 5, M0: 11) T3–4: 8 (N0: 4, N+: 4, M0: 8)	S: 4, S+RT: 7 S+RT: 3, S+CRT: 5	P: 1 (HPV16) N: 18	P: 1 N: 1 N/A: 17
Floor of mouth (C04)	13	49 (25–82)	M: 7 F: 6	Yes: 4 No: 5 N/A: 4	T1–2: 8 (N0: 6, N+: 2, M0: 8) T3–4: 5 (N0:2, N+: 3, M0: 5)	S: 5, S+RT: 3 S+RT: 1, S+CRT: 4	P: 0 N: 13	N/A: 13
Hard palate (C05.0)	3	66 (37–69)	M: 2 F: 1	Yes: 1 No: 0 N/A: 2	T1–2: 1 (N0: 1, N+: 0, M0: 1) T3–4: 2 (N0: 2, N+: 0, M0: 2)	S+RT: 1 S+RT: 1, S+CRT: 1	P: 0 N: 3	P: 0 N: 0 N/A: 3
All cases	285	57 (22–90)	M: 165 F: 120	Yes: 117 No: 71 N/A: 97	T1–2: 179 (N0: 107, N+: 72, M0: 179) T3–4: 106 (N0: 41, N+: 65, M0: 106)	S: 77, CRT: 1, S+RT: 82, S+CRT: 19 CRT: 9, S+RT: 22, S+CRT: 75	P: 10 (3.5%) N: 275 (96.5%)	P: 21 (42%) ** N: 29 (58 %) ** N/A: 238

* The smoking category also includes chewing tobacco mixture (Shamma). M: Male. F: Female. T1–2: Tumor size (T1 or T2). T3–4: Tumor size (T3 or T4). N: Lymph nodes. M: Metastasis (note that all M+ = 0). S: Surgery. S+RT: Surgery + radiotherapy. CRT: Chemo-radiotherapy. S+CRT: Surgery + chemo-radiotherapy. P: Positive. N: Negative. N/A: Not Available. ** Percentage out of 50 cases tested.

**Table 2 cancers-11-00820-t002:** Multivariate analysis using a Cox proportional hazard model to test the influence of various risk factors on overall survival in 285 patients with HNSCC.

Risk Factors	Categories	HR	95%CI	*p* Value
Age	Younger *	0.57	0.38–0.87	0.009
Gender	Females	1.01	0.66–1.55	0.963
Smoking	Non-smokers	0.77	0.48–1.22	0.258
Tumor site	Oropharynx	0.71	0.31–1.63	0.422
Tumor stage	Early (T1–2)	0.53	0.33–0.83	0.005
Treatment	Surgery **	0.40	0.20–0.77	0.006
HPV/p16 status	Positive	0.38	0.11–1.28	0.118

HR: Hazard Ratio. CI: Confidence Interval. * Younger denote patients whose age is ≤ the median age of 57 years/old. ** Surgery vs. any combined treatment.

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
