# Peer review of "Prevalence of Human Papillomavirus (HPV) Infection and the Association with Survival in Saudi Patients with Head and Neck Squamous Cell Carcinoma"

_cancers, 2019, doi:10.3390/cancers11060820_

Reviewer 1 Report

In this manuscript by Alsbeih and collaborators, the authors analyze the prevalence of HPV in a retrospective series of 285 HNSCC patients and show that oropharyngeal tumors are infrequent and that HPV-positivity is rare in this Saudi patient population. They confirm that, consistently with the literature, HPV and p16 positivity is associated with improved prognosis. Although the lack of statistical robustness of the study (only 10 HPV+ patients), which they acknowledge in the discussion section, the authors provide a manuscript which is clear and well written, and present a sound and well conducted study. Yet, some issues should be addressed for this manuscript to be suitable for publication.

Major points :

-The authors describe a close to 1:1 gender ratio (120 females for 165 males), which is rather unusual for HNSCC. It would be very interesting to discuss this observation in terms of epidemiology/risk factors in Saudi Arabia.

-Table 1 should include information about therapeutic options provided to patients. Is there a statistical difference of treatment provided to HPV-positive and HPV-negative patients?

-Treatments provided to the patients are potential confounding factors. They should therefore be included in the Cox model (Table 2).

-In the discussion session, the author state: “In an early global systematic meta-analysis that comprised 60 eligible studies and 182 included 5046 cases, the overall HPV prevalence in HNSCC was 25.9% with significantly higher 183 presence in oropharyngeal (35.6%) than in oral (23.5%) and in laryngeal (24%) cancers. In a more 184 recent meta-analysis, HPV DNA was detected in 32% (3837 out of 12163 cases) of HNSCC, with again 185 a higher prevalence in oropharyngeal (46%) than oral (24%) or laryngeal and hypopharyngeal (22%) 186 cancers”. However, the clinical relevance of HPV DNA positivity is debated, since they are likely to represent both transcriptionally  active (HPV RNA+) and inactive (HPV RNA-) HPV genomes. Detection of HPV RNA, which remains the gold standard for the detection of clinically relevant HPV+ lesions, is rare in non-oropharyngeal tumors (see  Combes JD & Franceschi S, 2014 Oral Oncol 50, 370-379). The authors use an algorithm that combine HPV DNA and p16 positivity, that is likely to more reliably estimate the proportion of HPV RNA+ tumors. Their results might therefore not be comparable to meta-analyses based on DNA detection only. In addition, the same issue can be highlighted when the authors present tumors with HPV-DNA and p16 positivity in their series as bona fide HPV+ tumors. They should acknowledge that, although being on the most reliable algorithm, this might identify false positives (HPV RNA- tumours).

-The authors conclude that : “Overall, these results imply that the prevalence 194 of HPV-driven HNSCC in our population is very low”. This conclusion might be biased by the fact that the proportion of oropharyngeal tumors is rather low in this series of patients (10% of the patients), HPV-positive lesions being extremely rare in non-oropharyngeal cancers.

Minor point:

In order to facilitate the comparison of their data with other studies, the authors could provide the CIMO code of tumor localization, based on the International Classification of Diseases for Oncology, if available.

Author Response

Reviewer 1:

Comments and Suggestions for Authors

In this manuscript by Alsbeih and collaborators, the authors analyze the prevalence of HPV in a retrospective series of 285 HNSCC patients and show that oropharyngeal tumors are infrequent and that HPV-positivity is rare in this Saudi patient population. They confirm that, consistently with the literature, HPV and p16 positivity is associated with improved prognosis. Although the lack of statistical robustness of the study (only 10 HPV+ patients), which they acknowledge in the discussion section, the authors provide a manuscript which is clear and well written, and present a sound and well conducted study. Yet, some issues should be addressed for this manuscript to be suitable for publication.

Answer: Thank you for your positive evaluation of the manuscript. The response to the issues raised are given below.

Major points :

1- The authors describe a close to 1:1 gender ratio (120 females for 165 males), which is rather unusual for HNSCC. It would be very interesting to discuss this observation in terms of epidemiology/risk factors in Saudi Arabia.

Answer: Thank for drawing our attention to this fact. The issue is now discussed in the revised version and a new reference was added in the revised manuscript in “Track Changes” mode, page 6, lines 196-200 as follow: “Females composed 42% of the patients compared to 58% of males in this cohort. This relatively high incidence in females is rather unusual for HNSCC; however, it confirms previous study in the country [28]. Although the exact reason is still unknown, it might be related to the increased females’ habits of consuming tobacco product (including Shamma) as 55% were smokers (68% in males)”.

2- Table 1 should include information about therapeutic options provided to patients. Is there a statistical difference of treatment provided to HPV-positive and HPV-negative patients?

Answer: Treatment options for patients, which essentially standardized and reflect tumor stage with some subtle adaptation to each individual case, were added to Table 1 in this revised version of the manuscript. Please see Table 1 in page 17 (Track Changes mode). Details of the treatment was already given in 4.2 of the manuscript.

For patients included in the study, HPV status was not taken into consideration in the treatment planning. Actually, the HPV status was not known for many patients before this study (with the exception of few cases with HPV/p16-related histopathologic features). Thus, treatment followed the same guideline for all patients with no difference between HPV-positive and -negative patients. This is now stated in the revised manuscript (Track Changes mode), page 10, lines 351-352 (Although some HPV-related histopathologic features were available for few cases, the treatment followed the same guideline for all patients with no difference between positive and negative HPV). Hopefully the results of this study will provide rational to personalize treatment according to HPV/p16 status.

3- Treatments provided to the patients are potential confounding factors. They should therefore be included in the Cox model (Table 2).

Answer: Treatment of patients is now included in the Cox PH model in Table 2 in the revised version. It essentially captures the results of the tumor stage. This is because the treatment was standardized according to stage with some subtle adaptation to each individual case. The result has also been discussed in the revised manuscript (Track Changes mode), page 8, lines 285-289, as follow: “In addition, the treatment offered to patients showed statistically significant (P = 0.006) association with OS. It also shows that surgery, which mainly underlies early stage tumors, result in higher survival compared to any other combined treatment. In other word, this result essentially captures that of the tumor stage since the treatment was stage standardized with some subtle adaptation to each individual case”

4- In the discussion session, the author state: “In an early global systematic meta-analysis that comprised 60 eligible studies and 182 included 5046 cases, the overall HPV prevalence in HNSCC was 25.9% with significantly higher 183 presence in oropharyngeal (35.6%) than in oral (23.5%) and in laryngeal (24%) cancers. In a more 184 recent meta-analysis, HPV DNA was detected in 32% (3837 out of 12163 cases) of HNSCC, with again 185 a higher prevalence in oropharyngeal (46%) than oral (24%) or laryngeal and hypopharyngeal (22%) 186 cancers”. However, the clinical relevance of HPV DNA positivity is debated, since they are likely to represent both transcriptionally  active (HPV RNA+) and inactive (HPV RNA-) HPV genomes. Detection of HPV RNA, which remains the gold standard for the detection of clinically relevant HPV+ lesions, is rare in non-oropharyngeal tumors (see  Combes JD & Franceschi S, 2014 Oral Oncol 50, 370-379). The authors use an algorithm that combine HPV DNA and p16 positivity, that is likely to more reliably estimate the proportion of HPV RNA+ tumors. Their results might therefore not be comparable to meta-analyses based on DNA detection only. In addition, the same issue can be highlighted when the authors present tumors with HPV-DNA and p16 positivity in their series as bona fide HPV+ tumors. They should acknowledge that, although being on the most reliable algorithm, this might identify false positives (HPV RNA- tumours).

Answer: We are grateful that you have raised this issue concerning HPV/DNA/RNA as related to p16 testing. We agree with the arguments that you have put forward. We have now revised the manuscript to include the following clarifications in the revised manuscript (Track Changes mode), pages 7-8 , lines 255-275: “It is known that the clinical relevance of HPV-DNA positivity is a matter of debate, because it is likely to represent both transcriptionally active (RNA+) and inactive (RNA-) HPV genomes. Therefore, detection of HPV-RNA by in situ hybridization is considered the gold standard for clinically relevant, HPV transcriptionally active lesions. However, the availability of this RNA methodology and concern for lower sensitivity compared to the affordable HPV-DNA polymerase chain reaction (PCR) led to the evaluation of IHC-p16 protein as a surrogate marker for the presence of active HPV in tumor cells [15]. Therefore, the UICC 8th edition defines HPV-mediated oropharyngeal cancer by use of p16 immunohistochemistry [36]. With this argument in mind, it is probable that the p16-positive samples in the 50 cases studied, represent active HPV infection, many of them could not be picked by the HPV-DNA PCR-based techniques. In such an arguable case, it is acknowledged that the incidence of HPV-related HNSCC would be higher (up to 42%) in our patients. However, larger study with more patients is needed to confirm this assumption, particularly that active HPV-RNA is considered rare in non-oropharyngeal tumors [37]. The alternative view could be that it is possible that p16 positivity is not exclusively related to HPV infection, which would debate its use as surrogate marker for the presence of HPV in all HNSCC [38]. Indeed, discrepancies in the p16/HPV-positivity have been observed and it is questionable if all HPV-positive and/or p16-positive tested cancers are HPV-driven. It is possible that sometimes HPV is an innocent bystander and p16 is independently positive [39]. This highlights the importance of identifying robust fingerprints of HPV-driven carcinogenesis to improve the estimate of HPV-attributable HNSCC and to predict the effectiveness of implementing preventive HPV vaccination and therapeutic interventions”.

5- The authors conclude that : “Overall, these results imply that the prevalence 194 of HPV-driven HNSCC in our population is very low”. This conclusion might be biased by the fact that the proportion of oropharyngeal tumors is rather low in this series of patients (10% of the patients), HPV-positive lesions being extremely rare in non-oropharyngeal cancers.

Answer: Our primary aim was to focus on oropharyngeal cancer; however, these turned out to be infrequent. Therefore, we have included oral cavity cancer, which include some anatomical sites (such as retromolar) susceptible to HPV-driven malignancies. The low incidence of oropharyngeal cancer found in this cohort is representative of data present at the national Saudi Cancer Registry and Globocan 2018 (please see lines 206-209). Although the incidence of HPV-DNA was overall low (3.5%) in our HNSCC patients, it was 21% in oropharyngeal cancer per se, which is not statistically different (P=0.16) from worldwide 36%-46% estimates. This is now made clearer in the revised manuscript. Please see abstract lines 23-24 (Overall, HPV-DNA was positive in 10 HNSCC cases (3.5%), mostly oropharyngeal (21%), and lines 30-31 (Although infrequent, oropharyngeal cancer increased over years and showed 21% HPV-DNA positivity, which is close to the worldwide 36%-46% estimates (P=0.16)); Results lines 144-145 (Thus, the highest HPV-DNA positivity was in oropharyngeal cancers (21%), which is not statistically different from the worldwide 36%-46% estimates (P = 0.16)); Discussion lines 231-232 (Although this 21% prevalence remains below the compiled worldwide estimates of 36%-46%, it was not statistically different (P = 0.16), indicating similar pathogenic association); Conclusion lines 433-434 (Although oropharyngeal cancer cases were infrequent, they increased over years and 21% were associated with HPV infection).

Minor point:

In order to facilitate the comparison of their data with other studies, the authors could provide the CIMO code of tumor localization, based on the International Classification of Diseases for Oncology, if available.

Answer: We have now included the ICD code of the International Classification of Diseases for Oncology, for each anatomical cancer site included in our study, in Table 1, to facilitate comparison of our data with other studies.

Finally, we thank you for your time and effort in reviewing the manuscript. We are grateful for your expert review and in-depth insights which have significantly improved the manuscript and enriched its scientific content. We hope that these modifications satisfy your requirements for publication.

Reviewer 2 Report

This paper describes the prevalence of HPV and p16 in a cohort of HNSCC patients in Saudi Arabia. The findings include low prevalence of HPV infection, with age, smoking, and tumor stage being associated with survival. 

My review of the manuscript includes the following critiques:

While the dates of diagnosis were stated in the methods section, it would have been better to include them easier in the paper, as well as to include an analysis of HNSCC cases over time; in many countries, the incidence of HPV-positive HNSCC is increasing, and despite the low numbers, would have been good to show that in this paper. 

While I do not discount the suitability of the HPV detection methods used, it is likely that microdissection of the FFPE sections to include DNA from exclusively the tumor areas may have increased the number of HPV-positive samples, particularly if the tumor was incredibly small. If the proportion of DNA from normal tissue compared to tumor is quite large, the sensitivity/specificity of any PCR-based test will be low.

Was betel nut chewing/quid included in the smoking category? If not, should it be? (unsure of the usage of this in the Saudi population). Were all/any tobacco products included in the smoking category? This should be clearly defined. 

Overall, a reasonable and mildly informative paper, would be very much improved with higher numbers for oropharynx, as it is difficult to draw any real conclusions with so few cases from that site.

Author Response

Reviewer 2:

Comments and Suggestions for Authors

This paper describes the prevalence of HPV and p16 in a cohort of HNSCC patients in Saudi Arabia. The findings include low prevalence of HPV infection, with age, smoking, and tumor stage being associated with survival.

Answer: Thank you for your positive evaluation of the manuscript. The response to the issues raised are given below.

My review of the manuscript includes the following critiques:

1. While the dates of diagnosis were stated in the methods section, it would have been better to include them easier in the paper, as well as to include an analysis of HNSCC cases over time; in many countries, the incidence of HPV-positive HNSCC is increasing, and despite the low numbers, would have been good to show that in this paper.

Answer: Thank you for bringing up this oropharyngeal cancer (OPC) and HPV incidence rates over years. Although the number of OPC and HPV+ were low, the data indicate increase in OPC and consequently HPV+ over years. In fact, OPC/HPV+ cases increased with time. By numbers, 8 OPC cases were diagnosed between 2002-2008 (142 patients) while 20 OPC cases were diagnosed between 2009-2016 (143 patients). Similarly, 2 HPV+ cases were diagnosed between 2002-2008 (142 patients) while 8 HPV+ cases were diagnosed between 2009-2016 (143 patients). This indicates that HPV-related HNSCC are increasing. This was added to the revised manuscript, please see:

Abstract lines 30-31: “Although infrequent, oropharyngeal cancer increased over years and showed 21% HPV-DNA positivity, which is close to the worldwide 36%-46% estimates (P=0.16)”.

Results, page 3, lines 100-101: “Interestingly, the number of oropharyngeal cancer cases increased with time, 8 cases were diagnosed in 2002-2008 (142 patients) compared to 20 cases in 2009-2016 (143 patients”.

Discussion, page 6-7, lines 209-210: “Nevertheless, our data indicate an increase in the number of oropharyngeal cancer cases over time as it doubled in the last decade in the country”.

Conclusion, page 11, lines 433-434: Although oropharyngeal cases were infrequent, they increased over years and 21% were associated with HPV infection”.

2. While I do not discount the suitability of the HPV detection methods used, it is likely that microdissection of the FFPE sections to include DNA from exclusively the tumor areas may have increased the number of HPV-positive samples, particularly if the tumor was incredibly small. If the proportion of DNA from normal tissue compared to tumor is quite large, the sensitivity/specificity of any PCR-based test will be low.

Answer: The collaborating pathologists had provided us with FFPE sections that are enriched with tumor samples. This is the regular and common procedures for this type of work to make sure that tumor tissues are abundant for the analysis. Therefore, the preponderant part of the extracted DNA was from tumor cells. In addition, we used 2 independent PCR-based techniques to detect and genotype HPV. Thus, we expect high sensitivity/specificity within the inherent limitation of each procedures. It is possible that p16 IHC is more sensitive as it picked more cases (42%) of supposedly transcriptionally active HPV, however it is less specific as we cannot exclude other mechanism leading to p16 overexpression. 

3. Was betel nut chewing/quid included in the smoking category? If not, should it be? (unsure of the usage of this in the Saudi population). Were all/any tobacco products included in the smoking category? This should be clearly defined.

Answer: betel nut is not known in Saudi Arabia (please see Reference #28: Hesham, et. al. Journal of International Oral Health 2017, 9, 105-109, doi:10.4103/jioh.jioh_58_17).

The dependency on any tobacco product consumption was included in the smoking category. This is now made clear in the footnote of Table 1 (*Smoking category includes also chewing tobacco mixture (Shamma). These are mainly cigarettes smoking and Shamma chewing, in addition to few Shisha (Hooka) users who were also cigarettes smokers.

4. Overall, a reasonable and mildly informative paper, would be very much improved with higher numbers for oropharynx, as it is difficult to draw any real conclusions with so few cases from that site.

Answer: The revised version has substantially improved the content of the manuscript. Although the number of OPC is not high, they showed HPV-DNA positivity (21%) that is close to the worldwide 36%-46% estimates (P=0.16). In addition, p16 positivity would suggest higher involvement. Therefore, we are planning to extend this study, include more OPC cases and systematically test for p16.

Finally, we thank you for your time and effort in reviewing the manuscript. We are grateful for your expert review and in-depth insights which have significantly improved the manuscript and enriched its scientific content. We hope that these modifications satisfy your requirements for publication.

Reviewer 3 Report

This manuscript reports the prevalence of HPV infection in a cohort of head and neck cancer patients in Saudi Arabia, and its association with clinical parameters. Findings reported here are interesting because describe for the first time the HPV burden in Saudi HNC patients that, interestingly, is significantly lower as compared to the overall worldwide estimate. Therefore, this work could open further researches to evaluate reasons beyond this discrepancy.

However, the conclusions are not fully supported by the presented results. In particular, the authors claim that the combined HPV/p16-positivity had better survival than HPV and/or p16 negative patients. Though, in figure 4C, HPV/p16 double positive patients showed a very similar survival curve as compared to HPV-/p16+ patients. Are these curves significantly different? Is the reported statistical significance relative to the overall differences between the three groups of patients? The authors should clarify this point and should report the Kaplan-Meier survival analysis by comparing only double-positive and single positive patients. 

In addition, the following minor points should be addressed:

-      Representative p16 negative and positive IHC staining should be reported.

-      Results showed in fig. 4 should be described in the results section and not in the discussion

Author Response

Reviewer 3:

Comments and Suggestions for Authors

This manuscript reports the prevalence of HPV infection in a cohort of head and neck cancer patients in Saudi Arabia, and its association with clinical parameters. Findings reported here are interesting because describe for the first time the HPV burden in Saudi HNC patients that, interestingly, is significantly lower as compared to the overall worldwide estimate. Therefore, this work could open further researches to evaluate reasons beyond this discrepancy.

Answer: Thank you for your positive evaluation of the manuscript. The response to the issues raised are given below.

1. However, the conclusions are not fully supported by the presented results. In particular, the authors claim that the combined HPV/p16-positivity had better survival than HPV and/or p16 negative patients. Though, in figure 4C, HPV/p16 double positive patients showed a very similar survival curve as compared to HPV-/p16+ patients. Are these curves significantly different? Is the reported statistical significance relative to the overall differences between the three groups of patients? The authors should clarify this point and should report the Kaplan-Meier survival analysis by comparing only double-positive and single positive patients.

Answer: Thank you for your comment and suggestion. The P-value reported represents the overall significance level. We made this clear in the revised version. Please see page 5-6, lines 145 (Overall, survival analysis showed a trend toward better OS for HPV-positive (67% survival) compared to HPV-negative (27% survival) that did not reach statistical significance (P = 0.12), due to most probably the small number of HPV-positive cases (Figure 4, panel A)).

In addition, we have conducted Kaplan-Meier analysis comparing HPV/p16 double-positive to single-positive which was not statistically significant (P = 0.85). This was reported in the revised manuscript page 5, lines 174-175 (However, there was no significant difference between double-positive and single positive patients (P = 0.85)), and legend to Figure 4, page 5, lines 153-155 (The P-value in panel C represents the overall significance level. There was no significant difference between single-positive and double-positive cases (P = 0.85)). This issue was also clarified in Figure 3, panel A. Please see legend to Figure 3, page 4 lines 131-133 (The P-value in panel A represents the overall significance level. Al pairwise comparisons were statistically significant (P <= 0.03) except T4 vs. T3 (P = 0.32).

Furthermore, we have corrected related statements in the whole manuscript:

Abstract lines 33-36 (The HPV and/or p16 positivity had better OS than HPV/p16 double negative patients (P=0.05). Thus, HPV/p16 status helps improve prognosis by distinguishing between the most favorable p16/HPV positive and the least favorable double-negative tumors).

Discussion, page 9, lines 312-318 (Interestingly, the Kaplan-Meier LogRank survival analysis showed a significant difference (P = 0.05) where HPV/p16-positive cases showed substantially better OS than double-negative patients. Although double-positive cases showed slightly better survival than single-positive patients, the difference was not statistically significant in this study, due to most probably the small number of patients who tested positive for HPV and/or p16).

Conclusion, page 11, lines 438-441 (These tests could improve the survival prediction by distinguishing between the more favorable HPV-positive/p16-positive and the less favorable double-negative HPV/p16 group of HNSCC patients whose have the worse prognosis).

2. In addition, the following minor points should be addressed:

2.1. -      Representative p16 negative and positive IHC staining should be reported.

Answer: Representative Histopathology and IHC example of p16 positive and negative staining was added to the manuscript as Figure 5. Please see revised manuscript page 5, line 163 (Examples of p16 protein IHC strong (positive) and weak (negative) staining is given in Figure 5), Figure 5, page 6, lines 177-182.

2.2.-      Results showed in fig. 4 should be described in the results section and not in the discussion

Answer: Results of Figure 4 has now been described in the result section under 2.2 (page 5, lines 145-148 (Overall, survival analysis showed a trend toward better OS for HPV-positive (67% survival) compared to HPV-negative (27% survival) that did not reach statistical significance (P = 0.12), due to most probably the small number of HPV-positive cases (Figure 4, panel A)); and 2.3 (page 5, lines 169-175 (Survival analysis showed a statistically significant (P = 0.02) better OS for p16-positive (64% survival) compared to p16-negative (29% survival) patients (Figure 4, panel B). In addition, the survival analysis of the combined HPV/p16 status in the 50 cases studied (Figure 4, panel C), showed an overall significant difference in OS (P = 0.05), whereby HPV and/or p16 positive patients displayed better survival (64%-67% survival) compared to HPV/p16 double positive patients (29% survival). However, there was no significant difference between double-positive and single positive patients (P = 0.85)). In addition, Figure 4 was moved up accordingly (page 5, lines 149-155).

Finally, we thank you for your time and effort in reviewing the manuscript. We are grateful for your expert review and in-depth insights which have significantly improved the manuscript and enriched its scientific content. We hope that these modifications satisfy your requirements for publication.

Round  2

Reviewer 1 Report

The authors provide satisfactory responses to the points I have raised. The manuscript is now suitable for publication.